# Bile Acids and Microbiota: Multifaceted and Versatile Regulators of the Liver–Gut Axis

**DOI:** 10.3390/ijms22031397

**Published:** 2021-01-30

**Authors:** Niklas Grüner, Jochen Mattner

**Affiliations:** 1Mikrobiologisches Institut-Klinische Mikrobiologie, Immunologie und Hygiene, Universitätsklinikum Erlangen and Friedrich-Alexander Universität (FAU) Erlangen-Nürnberg, 91054 Erlangen, Germany; Niklas.Gruener@extern.uk-erlangen.de; 2Medical Immunology Campus Erlangen, FAU Erlangen-Nürnberg, 91054 Erlangen, Germany

**Keywords:** bile acids, intestinal microbiota, host–microbe interactions, enterohepatic recirculation, microbial metabolism

## Abstract

After their synthesis from cholesterol in hepatic tissues, bile acids (BAs) are secreted into the intestinal lumen. Most BAs are subsequently re-absorbed in the terminal ileum and are transported back for recycling to the liver. Some of them, however, reach the colon and change their physicochemical properties upon modification by gut bacteria, and vice versa, BAs also shape the composition and function of the intestinal microbiota. This mutual interplay of both BAs and gut microbiota regulates many physiological processes, including the lipid, carbohydrate and energy metabolism of the host. Emerging evidence also implies an important role of this enterohepatic BA circuit in shaping mucosal colonization resistance as well as local and distant immune responses, tissue physiology and carcinogenesis. Subsequently, disrupted interactions of gut bacteria and BAs are associated with many disorders as diverse as *Clostridioides difficile* or *Salmonella* Typhimurium infection, inflammatory bowel disease, type 1 diabetes, asthma, metabolic syndrome, obesity, Parkinson’s disease, schizophrenia and epilepsy. As we cannot address all of these interesting underlying pathophysiologic mechanisms here, we summarize the current knowledge about the physiologic and pathogenic interplay of local site microbiota and the enterohepatic BA metabolism using a few selected examples of liver and gut diseases.

## 1. Introduction

The microenvironment, the tissue milieu and metabolic processes of the host’s own cells and colonizing microbial commensals have a tremendous impact on the development and function of various hematopoietic and non-hematopoietic cell populations in humans. The gut microbiota and their metabolites, for example, contribute to the regulation of many physiological processes and exhibit a pivotal role in maintaining human health. Thus, microbiota signal not only to neighboring but also to distant organs and tissues in the body and interfere, for example, with the brain (brain–gut axis), the liver (liver–gut axis) and the immune and hormone system of the host [1], and vice versa, the host also shapes the intraluminal microbiota. Although the genetic traits of the host as well as dietary and micro-environmental factors presumably contribute to these complex interactions [2,3,4,5,6], it is largely unknown how commensal microbiota are selectively influenced. 

Nonetheless, disrupted intestinal microbiota that accompany inflammatory processes in the gut and the subsequent altered availability of microbial- or host-derived products and metabolites are frequently associated with the pathogenesis of various complex disorders [7,8,9,10]. As the etiology of these diseases is multifactorial, they develop as a consequence of a malfunctioning network rather than of a single cause [11]. Thus, the versatile host–microbiota interactions are pivotal for maintaining physiologic homeostasis. 

One interesting aspect of host–microbiota interactions is that host- and microbiota-derived enzymes frequently regulate the same pool of intraluminal metabolites [12]. Some physiologic signaling cascades and metabolic processes even require co-metabolism between the host and the microbiota. A classic example of shared alternating enzymatic reactions between mammalians and their intraluminal microbiota is the biosynthesis of bile acids (BAs). In this case, the host synthesizes primary BAs in the liver, which the intestinal microbiota subsequently transform in the gut. This conjugation and deconjugation of BAs plays a pivotal role in many physiological and pathophysiological processes in the liver–gut axis. Here, we discuss a few of them using selected examples of infectious, immune-mediated and tumor diseases.

## 2. Bile Acids (BAs) and Enterohepatic Circulation

Bile acids (BAs) are hydroxylated, amphipathic steroid acids that are synthesized in the peroxisomes of the liver from cholesterol [13]. In hepatic tissues, BAs are also conjugated to the hydrophilic amino acids glycine (Gly), taurine (Tau) or sulfate. These BAs are now named primary or conjugated BAs. They consist of cholic acid (CA) and chenodeoxycholic acid (CDCA) or tauro- and glycoconjugated versions thereof (Table 1). While humans preferably use glycine for conjugation, taurine is the most common conjugate of BAs in rodents.

Once synthesized, these primary BAs are secreted into the bile, where they comprise about 80% of the organic compounds [29] and are concentrated for storage in the gallbladder [30]. Ingestion of food triggers the release of cholecystokinin by enteroendocrine cells, which causes gallbladder contraction and the release of primary BAs into the duodenum [31]. There, the primary purpose of BAs is to facilitate the digestion and absorption of dietary lipids, fatty acids, cholesterol, fat-soluble vitamins and other hydrophobic components of the diet via its surfactant properties, which emulsify fats into micelles [14]. Usually, more than 95% of the primary BAs are re-absorbed from the terminal ileum and transported back into the liver via the so-called enterohepatic circulation (EHC) (Figure 1). Upon their return to the liver via the portal vein, primary BAs inhibit cholesterol biosynthesis and further BA biosynthesis [29].

However, small quantities of primary BAs also reach the colon, where certain gut bacteria transform them into secondary BAs, such as deoxycholic acid (DCA), ursodeoxycholic (UDCA) and lithocholic acid (LCA), by deconjugation, oxidation/epimerization, (7-α-) dehydroxylation and esterification [32,33,34,35] (Table 2). In particular, secondary BAs exhibit strong antimicrobial activity and cytotoxicity. Thus, BAs also regulate the composition of gut bacterial communities and host physiology [32], features that are described in more detail below. Immunoglobulin A (IgA), the major immunoglobulin at mucosal surfaces, enhances the antimicrobial properties of BAs; thus, both BAs and IgA inhibit bacterial growth and adhesion and subsequently protect against ascending infections within the biliary tract. In addition, BAs eliminate bilirubin from the body via the feces [29,36].

Both, primary and secondary BAs exert biological effects on cells of the host by activating nuclear and plasma membrane receptors, including the nuclear farnesoid X receptor (FXR) or the G protein-coupled receptor (TGR5) [37,38,39,40]. These receptors control the synthesis and metabolism of BAs. However, engagement of these receptors enables BAs to contribute to the regulation of glucose homeostasis, lipid metabolism and energy expenditure [50]. Furthermore, BAs regulate immune responses upon ligation of these two receptors (Figure 2), which are located at the interface of the host immune system with the intestinal microbiota [51]. Both receptors are highly expressed on cells of the innate immune system, including macrophages, dendritic cells and natural killer T (NKT) cells [51]. In particular, TGR5 and FXR regulate the polarization of macrophages and can also rescue mice from severe colitis [52,53].

In summary, the main functions of BAs include (a) emulsifying and digesting fat, (b) regulating and excreting cholesterol, (c) exerting antimicrobial effects and (d) eliminating bilirubin from the body [29,36]. However, as there exist multiple microbes that are tolerant against bile [54], the antimicrobial effects of bile (acids) selectively restrain certain microbial species and subsequently affect the composition of the complete intestinal or biliary microflora.

## 3. Local Site Microbiota in the Gut and Bile

The intestine hosts complex and dynamic populations of highly diverse microorganisms. These include various bacteria, archaea and eukarya, which form a mutually beneficial relationship with the host [4,7]. Diet, for example, strongly influences the composition of the microbiota [55,56], and vice versa, intestinal microbiota produce metabolites and extract nutrients from a large range of molecules that enzymes of the host are unable to convert [34]. Many of these nutrients and metabolites derived from commensal microbiota have been implicated in the development, homeostasis and function of the immune system, suggesting that microbial commensals influence host immunity via nutrient- and metabolite-dependent mechanisms [57]. Accordingly, an altered composition of the gut microbiota, known as dysbiosis, accompanies many intestinal and extra-intestinal disorders [1,58].

Similar to the gut, the gallbladder also harbors a complex microbiota. In contrast to the intestine, the microbiota of the biliary tract contains relatively low levels of Bacteroidetes, while numbers of Proteobacteria, Tenericutes, Actinobacteria and Cyanobacteria are increased [59,60,61]. Similar to what is observed with intestinal microbiota, although less well studied, the bile microbiota can be disrupted [59].

## 4. Crosstalk between BAs and Microbiota

The regulation of the BA pool is one example of the interference of the microbial metabolism with the host [62]. The deconjugation, oxidation/epimerization, (7-α-) dehydroxylation and esterification of BAs by the intestinal microbiota can dramatically change their physicochemical properties and subsequently affect their microbial toxicity and intestinal absorption.

For example, the deconjugation of BAs by microbial bile salt hydrolases (BSH)—abundant enzymes found in all major bacterial phyla [33]—enhances their intestinal re-absorption. Furthermore, it promotes the colonization of the gut by microbiota and can serve as a nutritional source of sulfur, nitrogen and carbon [63,64,65].

7-α-Dehydroxylation converts the primary BAs cholic (CA) and chenodeoxycholic acids (CDCA) into the secondary BAs deoxycholic (DCA) and lithocholic acids (LCA) [35]. It is quantitatively the most important and the most physiologically significant conversion of BAs in humans. Deoxycholic acid may even account for up to 25% of the total BA pool. The bacterial species that possess 7-α-dehydroxylation activity include members of the Firmicutes phylum, such as Clostridium or Eubacterium.

The effects of the intestinal microbiota presumably extend further beyond BA composition and biotransformation. In fact, germ-free and antibiotic-treated mice exhibit reduced BA excretion in the feces. In contrast, the BA pool is increased along with enhanced BA secretion and re-absorption from the intestine and an overall altered metabolic homeostasis of the host [66,67,68,69].

As mentioned above, their amphipathic character also supplies BAs with antimicrobial activities. Thus, BAs alter the fluidity, permeability and function of cellular membranes and membrane-bound proteins [54,70,71]. BAs also cause DNA damage and oxidative stress and affect the formation of RNAs and proteins [54,70,71,72]. The application of BAs expands Firmicutes at the expense of Bacteroidetes in the gut. Accordingly, increased intraluminal BA concentrations favor the growth of bacterial species that 7-α-dehydroxylate primary BAs into secondary BAs [73,74]. In contrast, lower intraluminal BA levels predominantly favor the growth of Gram-negative bacteria. In addition, BAs also influence the integrity of intestinal epithelial cells and mucosal immune responses and thus indirectly regulate the composition of microbial communities [68,75,76,77]. As discussed below, secondary BAs can also exhibit toxic effects (Table 2) and promote infectious, inflammatory or malignant diseases.

## 5. Functional Consequences

A disruption of the versatile interactions between BAs and microbiota can lead to many disorders. However, it has remained unclear whether compositional changes in the intestinal microbiota and/or the intraluminal metabolome are the consequence or the cause of the respective individual diseases. Nonetheless, dysbiosis usually accompanies many disorders, such as inflammatory bowel disease (IBD), or forms the basis for infections with gastrointestinal pathogens. Below, we summarize a few examples of these diseases affecting the liver–gut axis.

### 5.1. Infectious Disease

Microbial metabolism affects BA composition. Microbiota and microbiota-derived products also confer colonization resistance against many gastrointestinal pathogens, and vice versa, BAs as well as dietary habits shape the composition of the microbiota [78,79,80]. The loss of secondary BAs, for example, has been associated with susceptibility to infection by pathogenic bacteria. Conversely, a restoration of the secondary BA pool again promotes colonization resistance [26].

Dietary fat, for example, boosts intraluminal primary BA concentrations and subsequently gut colonization by *Salmonella enteritidis serovar* Typhimurium (*S.* Typhimurium) [16], as *Salmonella* spp. adapt to bile [17], upregulate the expression of virulence genes [18] and thus exhibit a much higher bile resistance than intestinal microbiota [16]. Survival strategies utilized by *Salmonella* serovars include the upregulation of efflux pumps and outer membrane proteins, the modification of lipopolysaccharide (LPS) and membrane structures and the induction of many virulence factors, including the repression of the type 3-secretion system (T3SS), which is essential for bacterial invasion of intestinal epithelial cells [20]. This ability of *Salmonella* spp. to survive in bile is a longstanding observation that is utilized for their selection and isolation in selective media, such as the bile-salt containing MacConkey agar [81].

Both commensal and pathogenic *Escherichia coli* (*E. coli*) strains utilize several mechanisms to resist bile. These include the activation of stress response genes, promoting the repair of DNA or membrane damage, the induction of efflux pumps or the upregulation of toxin/antitoxin systems to remove bile compounds [19,20,21,22,23]. Pathogenic *E. coli* strains, including enteropathogenic *E. coli* (EPEC), enterotoxigenic *E. coli* (ETEC) and enterohemorrhagic *E. coli* (EHEC), also express virulence genes in response to bile exposure [20]. These primarily facilitate the colonization of the gut, the acquisition of nutrients and adhesion to host cells. However, further studies should explore whether there exists an altered resistance to detergents among different *E. coli* strains following bile exposure.

Exposure to bile in the small intestine also increases the virulence of *Shigella flexneri* and *Shigella dysenteriae*, two of the causative agents of Shigellosis [20], prior to reaching the site of infection in the colon. This manifests in enhanced protein secretion as well as an improved adherence to and invasion of intestinal epithelial cells.

Bile and the microbiota also influence the life cycle of *Clostridioides difficile* (*C. difficile*), a spore-forming Gram-positive bacterium and the causative agent of antibiotic-associated diarrhea (AAD). The application of antibiotics disrupts the commensal microbiota that converts primary BAs into secondary BAs, which usually inhibit the germination of *C. difficile* spores into vegetative bacteria [24,25,26]. Subsequently, the accumulation of primary BAs promotes the germination of *C. difficile* spores, bacterial replication and the production of colitis-mediating enterotoxins.

One of the few intestinal bacterial species that can actually convert primary BAs such as cholic acid into toxic secondary BAs such as deoxycholic acid is *Clostridium (C.) scindens* [82]. Due to the synthesis of *C. difficile*-inhibiting metabolites from host-derived bile salts, *C. scindens* enhances resistance to infection with *C. difficile* in a secondary BA-dependent fashion [83]. Thus, this BA-α-dehydroxylating member of the genus *Clostridium* is an example of a bile-mediated colonization resistance mechanism by intestinal microbiota. The knowledge of such mechanisms and the ecological context of microbes underlying these effects will facilitate the amplification of microbiota-mediated pathogen resistance in individuals at risk for infection.

Accumulating evidence demonstrates that intestinal microbiota also modulate enteric virus infections. For example, the modification of BAs by commensal bacteria primes type III interferon responses and subsequently inhibits an infection of the proximal small intestine with norovirus [84]. Primary and secondary BAs and their derivatives, such as glyco-ursodeoxycholic acid, and semi-synthetic derivatives, such as obeticholic acid, as well as ursodeoxycholic acid (UCDA), have been even recently suggested as therapy for SARS-CoV-2 infection due to the inhibition of virus binding to host cells or the suppression of the COVID-19-associated cytokine storm [85,86].

### 5.2. Inflammatory Bowel Disease (IBD)

Inflammatory bowel diseases (IBDs), such as ulcerative colitis (UC) and Crohn’s disease (CD), are characterized by immune-mediated inflammation within the gastrointestinal (GI) tract of affected patients [87]. Although the exact etiologies underlying UC and CD are unknown, both disorders are generally thought to result from a complex interplay of microbial, genetic, geographic and habitual factors [88,89], resulting in a dys-balanced interaction of symbiotic microorganisms, the intestinal epithelium and the immune system [90].

The absorption of BAs in the intestine is predominantly impaired in both pre-clinical models of experimental enterocolitis and in human IBD patients [27]. In addition, fecal BA composition remains altered in IBD patients who do not sustain remission [91]. Furthermore, the expression of the apical sodium-dependent bile acid transporter (ASBT), the major transporter for the efficient uptake of BAs in the terminal ileum [92], is suppressed in various animal models [27,93,94,95]. Accordingly, particularly CD patients exhibit reduced ASBT expression [96,97]. The subsequent increase of the intraluminal BA pool frequently manifests in the form of diarrhea due to a spillover of BAs into the colon [27]. Importantly, the restoration of the intestinal BA pool not only ameliorates clinical symptoms but also increases the numbers of regulatory T cells expressing the transcription factor RORγ and ameliorates host susceptibility to inflammatory colitis [98]. Increased levels of secondary BAs have been also associated with remission in UC patients following fecal microbiota transplantation (FMT) [99]. Thus, in summary, secondary BAs are associated with remission and improved clinical outcome in IBD, presumably reflecting a richer and more diverse microbiota.

### 5.3. Immune-Mediated Disease of the Liver

Chronic cholestatic liver diseases, such as primary biliary cirrhosis (PBC) and primary sclerosing cholangitis (PSC), are characterized by hepatic portal inflammation and slowly progress to obliterative fibrosis and eventually liver cirrhosis [100]. The subsequent obstruction of the bile flow alters the commensal microbiota of the gut and the biliary tract, susceptibility to infection and the integrity of the epithelial layers [61,101,102,103,104,105,106]. Indeed, increased concentrations of the pro-inflammatory and potentially cancerogenic agent taurolithocholic acid accompany biliary dysbiosis in PSC patients [101]. Furthermore, patients suffering from both PSC and IBD [107] exhibit distinct microbiota and microbiota-stool BA correlations as compared with IBD patients without concomitant PSC [108]. For example, the relative abundance of Clostridiales II and the overall bacterial diversity are lower in PSC patients compared with UC patients (without concomitant PSC) and control individuals [109]. Thus, microbial modifications of BAs likely modify the metabolism of the host, which can lead to altered immune signaling via BA receptors and modified immune responses. As microbial triggers play a role in the pathogenesis of both disorders [110,111], the altered BA composition in PBC and PSC might even allow unusual bacteria to expand and/or even perpetuate ascending infections within the biliary tree. Thus, there exists compelling evidence that microbial agents and/or changes in the intestinal/biliary microbiota, as well as an altered metabolite profile, including BAs, are involved in the pathogenesis of PBC and PSC. As pathogenic bacteria might use the biliary tree as a route for infection, and as ursodeoxycholic acid (UCDA, one of the secondary bile acids) is the only FDA-approved drug currently available for the treatment of PBC, its mechanism of action might include antimicrobial effects, as recently shown for *C. difficile* infections [112]. Whether the application of antibiotics might be a novel and alternative therapeutic option for the treatment of these devastating diseases needs to be addressed in future studies.

### 5.4. Hepato-Intestinal Carcinogenesis

It is well-known that the intestinal microbiota influences the efficacy of immunotherapy against tumors [113,114,115,116,117]. Furthermore, there exists increasing evidence that BAs and the intestinal microbiota promote intestinal and hepatic carcinogenesis [118,119,120]. Frequently, members of the genus Clostridiales are involved in these processes, presumably due to their capacity to convert primary into secondary BAs through 7-α-dehydroxylation [82].

Cholecystectomy alters the bile flow into the intestine and the enterohepatic circulation of BAs and can cause intestinal microbial dysbiosis [121,122,123]. The subsequent dysfunction of BA metabolism presumably positions patients at an increased risk for colorectal cancer (CRC). Thus, a lack of distinct intestinal microbiota promoted an accumulation of secondary BAs in the intestinal lumen, which may be a leading cause for the incidence of CRC in cholecystectomized patients [121,122,123]. The subsequent absence of surfactant protein D, which is synthesized in the gallbladder, contributes to the manifestation of intestinal dysbiosis [124].

Secondary BA-induced dysbiosis due to other medical reasons also promoted intestinal carcinogenesis [125] and adenomatous polyps [126], the most common precursor to CRC. In this context, primary BAs significantly inhibit cell responses following exposure to *Bacteroides fragilis* toxin (BFT or fragilysin), a metalloprotease encoded by enterotoxigenic *Bacteroides fragilis* (ETBF) that leads to tumorigenesis in susceptible mice, and is enriched in the mucosa of CRC patients [15] (Figure 3).

*Clostridium butyricum*, a butyrate-producing probiotic, inhibits intestinal tumor development by modulating Wnt signaling and gut microbiota [127]. Butyrate inhibited deoxycholic-acid-resistant colonic cell proliferation via cell cycle arrest and apoptosis [128]. The expression of the farnesoid X receptor (FXR), for which BAs are endogenous ligands [37,38,39,40], also influenced the Wnt/β-catenin signaling pathway and correlated inversely with the CRC stage and the clinical outcome [129]. Similarly, systemic butyrate also limited the antitumor effects of checkpoint inhibition [130]. Whether this was due to altered BA metabolism is currently unknown.

Removal of *Clostridium* species due to the application of antibiotics such as vancomycin promoted an accumulation of the primary bile acid chenodeoxycholic acid (CDCA). CDCA triggered the production of the chemokine CXCL16 by liver sinusoidal endothelial cells, which subsequently led to an accumulation of natural killer T (NKT) cells, an innate (-like) lymphocyte population endowed with potent immunomodulatory properties [131], in the liver and improved antitumor surveillance [28] (Figure 4).

Accordingly, alterations in the composition of intestinal microbiota involved in bile acid metabolism have been linked to the progression of hepatocellular carcinoma in mice and in patients with nonalcoholic steatohepatitis (NASH) [132,133]. Sex-dependent differences in BA profiles and altered hepatic BA retention in response to cholestyramine, a BA sequestrant, might thereby contribute to the sex-based disparity in liver carcinogenesis [134,135]. Furthermore, the deletion of thymine DNA glycosylase (TDG), a base excision repair enzyme that plays an essential role in the maintenance of epigenetic stability in cells [136], 8 weeks post-partum promoted hepatocellular carcinoma and dysregulation of BA homeostasis [119]. Future studies should investigate whether BA profiles and/or dysbiosis also affect pancreatic tumors.

## 6. Conclusions

There exists compelling evidence that co-metabolism between the host and the microbiota, as exemplified here for the enterohepatic circulation of BAs, is involved in the pathogenesis of many disorders of the gut–liver axis. Thus, the application of antibiotics to selectively deplete distinct microbiota or of BA sequestrants might be an alternative therapeutic option for the treatment of these complex disorders. However, many antibiotics are not specific for distinct bacterial species [137,138], similar to BA sequestrants for BAs. Thus, the choice of the substance, the duration of its use, its bile-permeability and its availability need to be carefully evaluated, especially with respect to the fact that many of the microbial species and their function for the maintenance of immune tolerance have not yet been elucidated. In order to address these pending questions, clinical studies must be initiated. Thus, therapeutic regimens need to be carefully considered in order to identify therapeutic targets and approaches that can be used to guide the development of effective therapies for dysbiosis and the disruption of BA metabolism, but also allow for the identification of common targets in enterohepatic diseases for clinical intervention in the future.

## Figures and Tables

**Figure 1 ijms-22-01397-f001:**
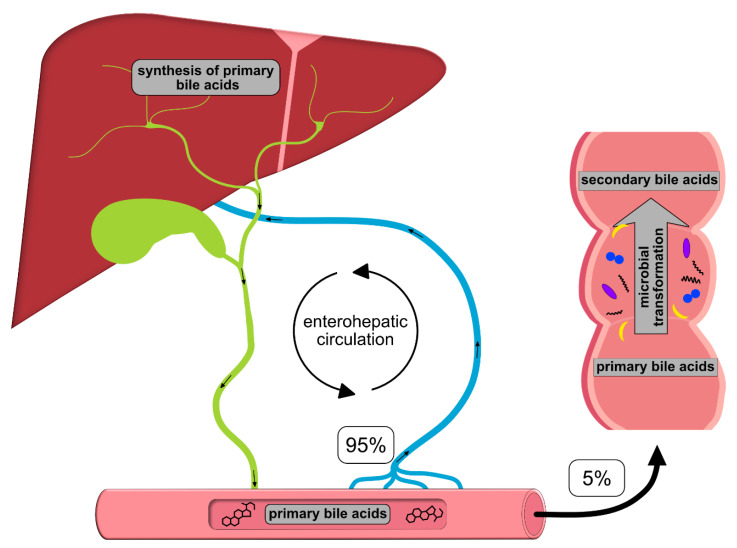
Schematic overview of the enterohepatic circulation (EHC). Primary bile acids (BAs) are synthesized in the liver and secreted into the duodenum with the bile; 95% of the BAs are re-absorbed in the terminal ileum and transported back to the liver for recycling. The remaining 5% enter the colon, where they are transformed into secondary BAs by colonic microbiota.

**Figure 2 ijms-22-01397-f002:**
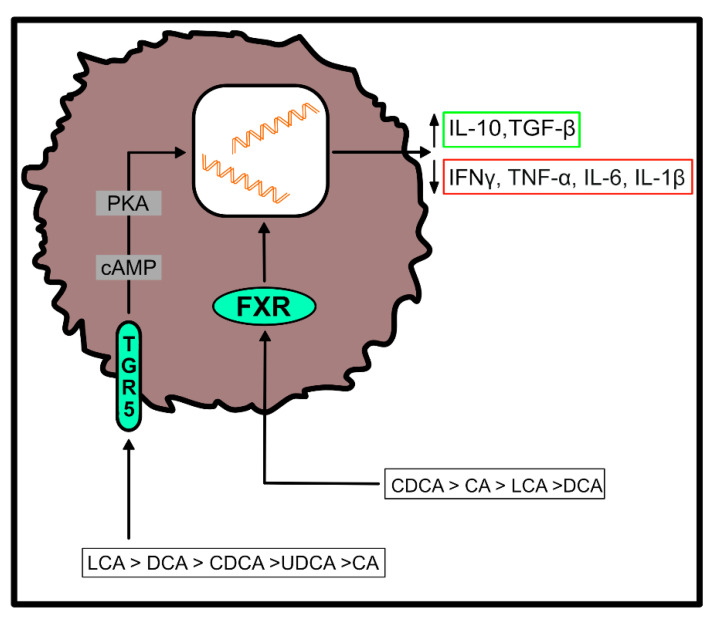
Effects of bile acid (BA) receptor engagement on immune cell function. After binding to the farnesoid X receptor (FXR) or the G protein-coupled receptor (TGR5), primary and secondary BAs shift the cytokine profile of myeloid cells to an anti-inflammatory phenotype. Their agonistic functions thereby depend on the affinity of individual BAs for the respective receptor, as indicated in the figure [51]. (IL-10 = interleukin-10; TGF-β = transforming growth factor beta; IFNγ = interferon gamma; TNF-α = tumor necrosis factor alpha; IL-6 = interleukin-6; IL-1β = interleukin-1 beta).

**Figure 3 ijms-22-01397-f003:**
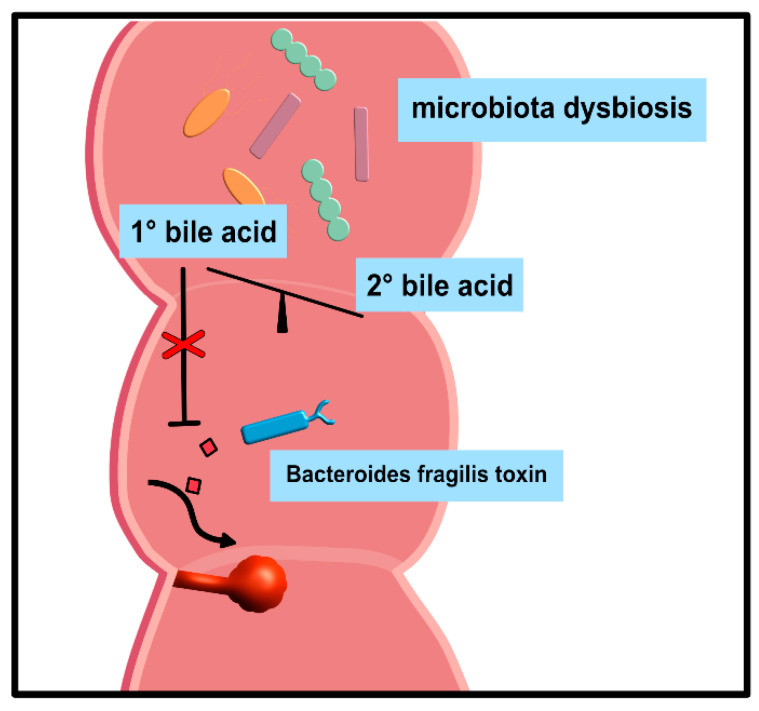
Influence of bile acid (BA) composition on colonic cell responses to the toxin of *Bacteroides fragilis*. The disruption of intestinal microbiota reduces the accumulation of primary BAs in the gut. Subsequently, the protective effect of primary BAs on colonic cell responses is lost and renders these cells susceptible to damage upon exposure to the toxin of *Bacteroides fragilis*.

**Figure 4 ijms-22-01397-f004:**
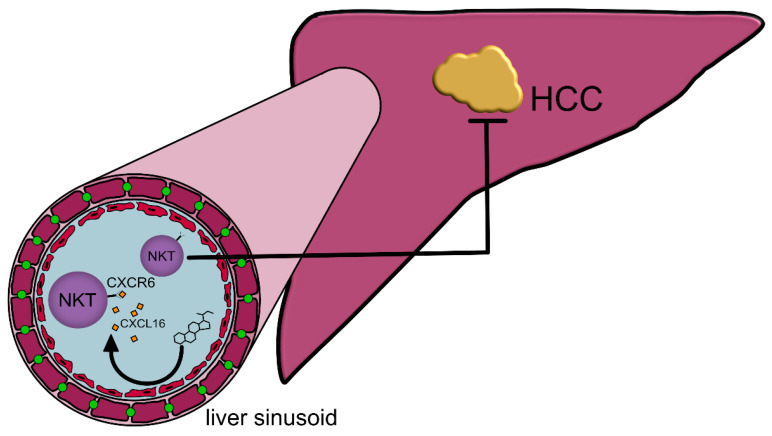
Inhibition of liver cell growth through natural killer T (NKT) cells. Chenodeoxycholic acid (CDCA) enhances the expression of CXCL16 on liver sinusoidal endothelial cells. Subsequently, CXCR6^+^-NKT-cells accumulate in the liver and protect against hepatocellular carcinoma (HCC).

**Table 1 ijms-22-01397-t001:** Primary bile acids and their functions under physiological and pathophysiological conditions. (UC = ulcerative colitis, CRC = colorectal cancer, HCC = hepatocellular carcinoma).

Abbrevation	Bile Acid Name	Physiologic Functions	Pathophysiologic Functions
CA	Cholic acid	facilitate digestion [14]; emulsify hydrophobic food components such as fats into micelles for intestinal absorption [14]; protect from CRC induced by *Bacteroides fragilis* toxin [15]	favor infections with pathogenic bacteria (*Salmonella* spp., *E. coli*, *Shigella* *dysenteriae*) [16,17,18,19,20,21,22,23] and the germination of *C. difficile* spores [24,25,26]; can cause colitis and diarrhea in UC patients [27]; promote the accumu-lation of CXCR6^+^-NKT-cells in the liver and protects from HCC [28]
CDCA	Chenodeoxycholic acid		
GCA	Glycocholic acid		
GCDCA	Glycochenodeoxycholic acid		
TCA	Taurocholic acid		
TCDCA	Taurochenodeoxycholic acid		

**Table 2 ijms-22-01397-t002:** Secondary bile acids and their (patho-)physiologic functions (CRC = colorectal cancer, PBS = primary biliary cirrhosis, ILC3 = type 3 innate lymphoid cells, PSC = primary sclerosing cholangitis, NASH = non-alcoholic steatohepatitis).

Abbrevation	Bile Acid Name	Physiologic Functions	Pathophysiologic Functions
DCA	Deoxycholic acid	exhibit antimicrobial activity [32]; maintain colonic microbiota [32]; perpetuate endocrine functions via binding to nuclear factor X receptor (FXR) and G-protein-coupled bile acid receptor (TGR5) [37,38,39,40]	enhanced levels are associated with CRC development [41]
UDCA	Ursodeoxycholic acid		drug for the treatment of primary biliary cirrhosis (PBC)
HDCA	Hyodeoxycholic acid		suppresses intestinal cell proliferation and enhances abundance of microbiota [42]
GDCA	Glycodeoxycholic acid		induces interleukin-22 production by ILC3s and improves ovulatory dysfunction and insulin resistance in patients suffering from polycystic ovary syndrome (POCS) [43]
GUDCA	Glycoursodeoxycholic acid		neuroprotective agent (https://pubchem.ncbi.nlm.nih.gov/compound/Glycoursodeoxycholic-acid)
LCA	Lithocholic acid		enhanced levels in patients suffering from Alzheimer´s disease [44]
GLCA	Glycolithocholic acid		
TLCA	Taurolithocholic acid		induces biliary dysbiosis in PSC patients
TDCA	Taurodeoxycholic acid		
TUDCA	Tauroursodeoxycholic acid		protective effects in NASH [45]; associated with attenuated hepatocarcinogenesis [46]
THDCA	Taurohyodeoxycholic acid		protective effects in experimentally induced colitis [47]
TMCA (a+b)	Tauromuricholic acid (alpha + beta)		maintains lipid and glucose metabolism [48]; reduced levels in the plasma of APP/PS1 mice, which are a mouse model for Alzheimer’s disease [49]

## Data Availability

We adhere to the Code of Conduct of the Committee on Publication Ethics (COPE) and to its Best Practice Guidelines.

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
