# Peer review of "Bile Acids and Microbiota: Multifaceted and Versatile Regulators of the Liver–Gut Axis"

_ijms, 2021, doi:10.3390/ijms22031397_

Round 1

Reviewer 1 Report

This is an interesting review on the interaction between bile acids and the microbiota and how dysbiosis can affect the production of secondary bile salts that may be associated with infectious and inflammatory pathologies and with intestinal and hepatic carcinogenesis.

The authors describe the synthesis of bile salts and enterohepatic circulation, the interaction between microbiota and bile salts to produce secondary bile salts and, the end they review some evidence that secondary bile salts produced by an anormal microbiota correlated with certain pathologies. The manuscript is well structured and developed, but it needs some small changes that improve its understanding.

General comment:

The reader, in the first part of the review is left with the idea that secondary BAs have always beneficial effects for the host, e.g., they have anti-inflammatory effects, as shown in Figure 2. But as discuses later in the manuscript, depending bacteria type, especially when there is dysbiosis, toxic secondary BAs can be form and promote infections, inflammation, and cancer. This could be introduced in section 4 in order to better position the reader. It could also be very enlightening to add a table with the most abundant primary BAs and some of their secondary BAs, depending on phylum and/or pathogens and relate them to beneficial or toxic effects. The table could be useful to introduce the BAs abbreviations that appear in the text. Since the subject is complex this table could help the reader a lot.

Specific comments:

- With the exception of Figure 1, none of the other figures have a title. A title must be added to each figure to situate the reader.

- Figure 1: In the legend there are abbreviations that do not appear in the figure! It would be necessary to put "primary bile acids" in liver figure where they are synthesized.

- Overall, it requires more uniformity in the use of abbreviations in the text and figures. For example, in Figure 2 there are some abbreviations that have not been introduced before, they should be specified in the figure legend. In addition, in the figure 2 also does not mentioned that is refers to secondary bile salts. It is necessary to modify the Figure 2 legend to make it clearer.

- Figure 3A does not exists in the manuscript but is cited in the text (line 383). Correct the mistake.

- Figure 4: the abbreviation HCC is not specified in the legend.

Author Response

The authors would like to thank the referees for their appreciation of the importance of our work and for their helpful and constructive critique. Below please find a point-by-point reply to the questions raised by the reviewers. The replies are written in italics underneath the respective questions.

The respective changes made in the manuscript are highlighted in tracker.

Reviewer 1:

This is an interesting review on the interaction between bile acids and the microbiota and how dysbiosis can affect the production of secondary bile salts that may be associated with infectious and inflammatory pathologies and with intestinal and hepatic carcinogenesis.

The authors describe the synthesis of bile salts and enterohepatic circulation, the interaction between microbiota and bile salts to produce secondary bile salts and, the end they review some evidence that secondary bile salts produced by an anormal microbiota correlated with certain pathologies. The manuscript is well structured and developed, but it needs some small changes that improve its understanding.

General comment:

The reader, in the first part of the review is left with the idea that secondary BAs have always beneficial effects for the host, e.g., they have anti-inflammatory effects, as shown in Figure 2. But as discuses later in the manuscript, depending bacteria type, especially when there is dysbiosis, toxic secondary BAs can be form and promote infections, inflammation, and cancer. This could be introduced in section 4 in order to better position the reader. It could also be very enlightening to add a table with the most abundant primary BAs and some of their secondary BAs, depending on phylum and/or pathogens and relate them to beneficial or toxic effects. The table could be useful to introduce the BAs abbreviations that appear in the text. Since the subject is complex this table could help the reader a lot.

At the end of section 4 in lines 178-180 we refer now to the pathologic effects of secondary bile acids that we will outline in some examples in section 5.

Two tables (table 1 +2) with the different bile acids and their (patho-) physiological functions have also been added. The abbreviations of the BAs are now also introduced at the respective sites in the text.

Specific comments:

- With the exception of Figure 1, none of the other figures have a title. A title must be added to each figure to situate the reader.

Respective titels to all figure legends have now been added.

- Figure 1: In the legend there are abbreviations that do not appear in the figure! It would be necessary to put "primary bile acids" in liver figure where they are synthesized.

This has been corrected, accordingly.

- Overall, it requires more uniformity in the use of abbreviations in the text and figures. For example, in Figure 2 there are some abbreviations that have not been introduced before, they should be specified in the figure legend. In addition, in the figure 2 also does not mentioned that is refers to secondary bile salts. It is necessary to modify the Figure 2 legend to make it clearer.

The abbrevations have been specified now in the figure legends to figure 2 and in the list of abbrevations at the end of the text. Since CA and CDCA are primary bile acids, we refer now to primary and secondary bile acids in this figure legend.

- Figure 3A does not exists in the manuscript but is cited in the text (line 383). Correct the mistake.

We refer now to Figure 3 in line 316.

- Figure 4: the abbreviation HCC is not specified in the legend.

The abbreviation is now specified in the figure legend and in the list of abbrevations at the end of the text.

Reviewer 2 Report

This paper was well written and I beleive it will be interested by readers. 

Author Response

The authors would like to thank the referees for their appreciation of the importance of our work and for their helpful and constructive critique. Below please find a point-by-point reply to the questions raised by the reviewers. The replies are written in italics underneath the respective questions.

The respective changes made in the manuscript are highlighted in tracker.

Reviewer 2:

This paper was well written and I beleive it will be interested by readers. 

Thank you very much for this comment and the great rating.  

Reviewer 3 Report

Authors of the article entitled "Bile acids and microbiota – multifaceted and versatile regulators of the liver - gut axis" decided to review scientific literature on topic of correlations between gut microbiota, bile acids, health and pathophysiology of certain diseases in humans.

Broad comments

Presented manuscript is well organized, in comprehensive way reviews many interesting aspects of gut-liver axis together with explaining ways in which bile acids can affect gut microbiota and hosts' cells. Authors included evaluation of well-known physiological functions of BAs together with current, on-point results from recent publications. Overall, the reception of presented article is good, but there are a few aspects that should be improved including editing and technical aspects of the manuscript.

Specific comments

  1. Authors reviews broad spectrum of scientific literature. Presentation of data showing potential benefits of BAs in treatment of COVID-19 is a great additional asset to this study.
  2.  As i understand from this review primary bile acids can help growth of pathogenic bacteria, and secondary bile acids present animicrobial properties. However on the end of the article primary bile acids are shown as a group preventing from carcenogenesis and secondary seem to promote this process. Why is that? What would expain this counter function in these two aspects. Maybe a table comparing primary and secondary bile acids in physiology and pathophysiology would be helpful to show these differences. 
  3. Authors might look for articles associating cholecystectomy with changes in gut microbiota composition and changes in BAs metabolism. That would be an interesting aspect worth reviewing. 
  4. Authors should expand 2 paragraph and write in 1-2 sentences about primary bile acids together with their abbreviations, because in lines 81-83 secondary bile acids are expained, and before that there is no information about cholic and chenodeoxycholic acid. It would be more undenstandable for the readers if authors first wrote about primary acid and then focus on secondary acids. 
  5. Authors should recheck abbeviations in the text, because for example
    chenodeoxycholic acid was first mentioned in line line 139, and its abbreviation was written in line 298,
    HCC is not explained in the description of Figure 4.
    Overall, list of abbreviations [line 339] does not show all abbreviations from the manuscript. 
  6. Editing errors:
    Line 100 - after severe colitis in blank space with citation, something is missing.
    Line 145 - there are 2 times "extend" in this line
    Line 283 - there is no Figure 3A in the manuscript.

Author Response

The authors would like to thank the referees for their appreciation of the importance of our work and for their helpful and constructive critique. Below please find a point-by-point reply to the questions raised by the reviewers. The replies are written in italics underneath the respective questions.

The respective changes made in the manuscript are highlighted in tracker.

Reviewer 3:

Authors of the article entitled "Bile acids and microbiota – multifaceted and versatile regulators of the liver - gut axis" decided to review scientific literature on topic of correlations between gut microbiota, bile acids, health and pathophysiology of certain diseases in humans.

Broad comments

Presented manuscript is well organized, in comprehensive way reviews many interesting aspects of gut-liver axis together with explaining ways in which bile acids can affect gut microbiota and hosts' cells. Authors included evaluation of well-known physiological functions of BAs together with current, on-point results from recent publications. Overall, the reception of presented article is good, but there are a few aspects that should be improved including editing and technical aspects of the manuscript.

Specific comments

  1. Authors reviews broad spectrum of scientific literature. Presentation of data showing potential benefits of BAs in treatment of COVID-19 is a great additional asset to this study.

Thank you very much for this comment.

  1. As i understand from this review primary bile acids can help growth of pathogenic bacteria, and secondary bile acids present animicrobial properties. However on the end of the article primary bile acids are shown as a group preventing from carcenogenesis and secondary seem to promote this process. Why is that? What would expain this counter function in these two aspects. Maybe a table comparing primary and secondary bile acids in physiology and pathophysiology would be helpful to show these differences. 

Two tables (table 1 + 2) with the different bile acids and their (patho-) physiological functions have now been added. At the end of section 4 in lines 178-180 we refer now to the pathologic effects of secondary bile acids that we will outline in some examples in section 5.

  1. Authors might look for articles associating cholecystectomy with changes in gut microbiota composition and changes in BAs metabolism. That would be an interesting aspect worth reviewing. 

We added a respective paragraph to chapter 5.4, lines 300-307 and discuss the effect of cholecystectomy in the context of intestinal carcinogenesis.

  1. Authors should expand 2 paragraph and write in 1-2 sentences about primary bile acids together with their abbreviations, because in lines 81-83 secondary bile acids are expained, and before that there is no information about cholic and chenodeoxycholic acid. It would be more undenstandable for the readers if authors first wrote about primary acid and then focus on secondary acids.

We introduce now cholic acid (CA) and chenodeoxycholic acid (CDCA) and the tauro- and glycoconjugated versions of both primary BAs in lines 60-62 and refer in this context now also the newly introduced table 1.

  1. Authors should recheck abbeviations in the text, because for example
    chenodeoxycholic acid was first mentioned in line line 139, and its abbreviation was written in line 298,

Abbrevations have been re-checked. A new and extended list of abbrevations has also been added at the end of the text.
HCC is not explained in the description of Figure 4.

The abbreviation is now specified in the figure legend and in the list of abbrevations at the end of the text.

Overall, list of abbreviations [line 339] does not show all abbreviations from the manuscript. 

The list with abbrevations has been extended, accordingly.

  1. Editing errors:
    Line 100 - after severe colitis in blank space with citation, something is missing.

The citations have not been properly connected. There is no text missing.  

Line 145 - there are 2 times "extend" in this line

This has been corrected.

Line 283 - there is no Figure 3A in the manuscript.

We refer now to Figure 3 in line 316.

Round 2

Reviewer 1 Report

After the revision, the authors provide an improved version incorporating the suggested changes.

The updated new version is now acceptable for publication.

Only two suggestions:

- Tables 1 and 2 would be better if the column “bile acid type" was removed since this is clearly indicated in the title of the table.

- Title of figure 4 need add ".......by Chenodeoxycholic acid"